# The Cohabitation of Humans and Urban Cats in the Anthropocene: The Clash of Welfare Concepts

**DOI:** 10.3390/ani11030705

**Published:** 2021-03-05

**Authors:** Filip Jaroš

**Affiliations:** Department of Philosophy and Social Studies, University of Hradec Králové, 50003 Hradec Králové, Czech Republic; filip.jaros@uhk.cz

**Keywords:** domestic cat, animal welfare, feral cats, pets, trap-neuter-return, routine neutering, population dynamics

## Abstract

**Simple Summary:**

Current cultural shifts in Western countries have changed the position of the cat to a companion animal, and its traditional role as a pest controller is no longer recognized by city dwellers. In a growing number of theoretical and field studies, the hunting abilities of cats and their high fertility are perceived as environmental risks. Bringing together theoretical perspectives from human–animal studies, animal ethics, population ecology, and biosemiotics, I highlight the existence of two different ecological (and even cultural) communities inhabiting urban environments: the culture of feral cats and the humano–cat culture of pets. Arguments are given for the essential role of feral cats in the population dynamics of the species when a growing number of pet cats are routinely neutered. Whereas neutering is presented by animal shelters and veterinary institutions as a universal means for improving cat welfare, it is at odds with the psychobiological needs of cats as viewed by a laissez-faire approach. This leads us to the conclusion that instead of one type of management of free-roaming cats, individual solutions should be sought to achieve a balance between the welfare of cats, other species, and human cultures in diverse urban environments.

**Abstract:**

Urban environments are inhabited by several types of feline populations, which we can differentiate as feral cats, free-roaming pets, and confined pets. Due to a shift in the cultural representation of cats from pest controllers to companion animals, cats living semi-independently of humans are perceived increasingly negatively, while the pet population has become the object of intense care. A regulative approach converges with a concern for welfare in the operation and educational campaigns of municipal shelters, which through their implementation of neutering policies have proven to be key players in the contemporary relation of urban cats and humans. The generally widespread notion of cat welfare associated with a secure life comes into tension with the fact that the psychobiological needs of feral cats are significantly different than those of pets. It becomes apparent that individual interactions between humans and cats in urban environments in the Anthropocene are increasingly influenced by the intervention of institutions that can be characterized as seeking to administer the wild.

## 1. Introduction

The cohabitation of cats, humans, and other species in urban residences represents a complex social, ecological, and, increasingly, ethical problem. A growing number of studies portray cats as dangerous predators that threaten the stability of bird, small mammal, and reptile populations or as carriers of dangerous diseases. A different perspective sees cats (along with dogs) as the most favored household pets, and concern for the welfare of cats motivates the actions of many individuals and municipal institutions. Cats occupy a dual role as “autonomous predator and ostensibly dependent companion” [1]. This deeply ambivalent attitude toward cats is inevitably mirrored in the problematic practice of trying to regulate the cohabitation of humans and cats in particular towns. This ambivalence is often to be found in the thinking of those who associate a negative image of cats with feral colonies yet reserve a positive one for individual cats who have owners.

The Anthropocene epoch can be characterized not only by the increased impact of human activity on the biosphere, but also by the greater determination of people to regulate the ecological relations of species (e.g., conservation programs, intervention against invasive species). Associated with this is a concern for animal welfare, extending ethical considerations from humans to include other animal species. In the case of cats living in urban environments, there arises the question of how to harmonize the interests of individual groups of residents, cats, and the species cats prey upon. Theoretical studies usually assume that from an environmental perspective, cats are an alien species that threatens populations of small vertebrates due to their exceptional predatory skills [2,3]. Their discourse repeats the problems of invasive species ecology [4].

From the point of view taken in the present study, this view is problematic for at least two reasons: (i) in regions where domestic cats have been living for several centuries (particularly the “Old World”), they can rightly be considered a natural part of the ecosystem, since there prevails “a high degree of adaptation of local wildlife to cats” [5]; (ii) urban areas are characterized by a high concentration of both cat and bird populations [6,7]; (iii) domestic cats ecologically compete with/prey on the predators of bird nests like the brown rat (*Rattus norvegicus*), beech marten (*Martes foina*), and garden dormouse (*Eliomys quercinus*). Domestic cats are generalist and obligate predators that receive supplementary food, and their population density reflects that of humans more than the density of their prey [8]. In urban ecology, the classical distinction of nature and culture is problematic: rather than “wildlife,” in urban areas it is more accurate to speak of synanthropic species that have inhabited city spaces at different times. From this perspective, the common blackbird (*Turdus merula*), for example, is a more recent arrival in urban spaces than cats. As this commentary concerns the problem of cats in urban environments, it does not take into account cases of wild populations threatening endemic species (particularly in fragile island ecologies) or farm cats occasionally preying on wildlife. Therefore, it is justified (particularly in the Old World) to view free-roaming cats as a natural part of the character of urban areas rather than an invasive species [9,10,11,12].

Free-ranging cats are individuals with the characteristics of semi-wild commensal animals whose important ecological and ethological functions, unlike fully domesticated animals, are not under human control [13]. The legal position in many European countries that defines a cat as a domestic animal and presupposes a distinct owner for each individual thus does not reflect the variability of ecological and social niches in this species [14]. Feral colonies can receive supplemental feeding from cat lovers, but “feeding ladies” are not responsible for the behavior of the individuals they assist. Furthermore, the application of a given welfare concept has different impacts on individuals from different cat populations. For feral cats, veterinary care means removal from its environment and the endurance of significant stress, as they perceive humans primarily as predators, whereas free-ranging pets are acculturated to a degree that a visit to the veterinarian does not represent a decisive intervention in their lives (with the exception of neutering). The situation is further complicated by the fact that animal welfare can be evaluated according to three criteria: (i) affective states, (ii) natural living, (iii) basic health and functioning. Animal welfare involves different components that can be grouped roughly under these headings, which involve considerable but imperfect overlap. It is crucial to understand that the pursuit of any one criterion does not guarantee a high level of welfare as judged by the others [15].

For the sake of simplicity, I will group criteria (i) and (ii) in the case of cats under the laissez-faire policy that allows them to fulfill their psychobiological needs, while the veterinary view (iii) is based on the utilitarian perfectionist stance (more on this in Section 2) [16]. While the laissez-faire policy is applied in the professional literature primarily to members of wild species [17], the veterinary view prevails in the approach to the welfare of domestic animals, including cats [18]. It is important to note, however, that the first view is very widespread among cat owners, particularly in rural areas and in poorer urban areas [19]. In the approach to cats living in urban areas, we also see an intense clash of welfare concepts, expressed most forcefully around the issues of the free movement and reproductive possibilities of pets. I aim to demonstrate that the universal labeling of owners with a more liberal approach as “irresponsible” is the result of an excessive simplification of a complex issue. Key to this study is an understanding that proper management of urban cats is not a matter only of scientific facts, but also of cultural and ethical values manifested in the preferences of urban residents and in differences of welfare criteria. Every effort to adjust the relationship of humans and cats in urban environments is thus based on both scientific evidence and value assumptions that are of a different character in the case of feral cats, free-roaming pets, and confined pets, as will be demonstrated in Section 3 (threefold modeling). It will also be shown that the welfare of an individual does not necessarily overlap with the interests of the population to which it belongs—population genetics [20] and group-behavioral specifics (cultures) [21] must be considered here.

While conscious that circumstances may differ in individual countries, I proceed on the basis of studies carried out in Great Britain and the United States, which will be supplemented in places with the situation in other Western countries. The aim here is not a comparative study of the development of individual populations, but the utilization of empirical studies to identify the conflicts that arise when applying different approaches to cat welfare.

## 2. Urban Cat Populations as Distinctive Cultures

Drawing on the work of Natoli and Sandøe, I will differentiate the following three groups of synanthropic domestic cats (*Felis silvestris catus*) living in urban areas: *feral cats*, *free-roaming pets*, and *confined pets* [19,22,23]. Feral cats live in loose associations and can be found in public spaces. Urban environments provide them suitable shelter and sufficient nutrients, both from people (food scraps) and through predation (mostly rodents). While they are typically wary of people, they can develop relationships with specific people who feed them (known in the English literature as feeding ladies). Free-roaming pets include cats typically associated with a single household but which have the possibility of free movement in an urban environment. They come into contact not only with other pets, but occasionally with ferals as well. Their degree of dependence on a particular household varies from case to case, but it can generally be said that when appropriately cared for, they seek regular contact with their owner, in connection with the intake of the majority of their nutritional supply. Confined pets are tied to a specific household without the possibility of movement beyond the space of the house (with the possible exception of supervised movement, e.g., around a vacation home). They are fully dependent on people, who also make decisions on their reproductive possibilities. Reproduction is generally allowed only to pedigree cats, as with domestic short-haired cats the behavioral manifestations associated with intact individuals come into conflict with the restrictions of urban apartments. 

In practice, it is difficult to precisely distinguish individual categories of cats: “stray cats” (or also semi-feral) can be perceived as a transitional category between ferals and free-roaming pets. These individuals are not tied to any particular household but can receive supplemental feeding from residents of households and at the same time transiently join colonies of feral cats. For our purposes, however, it is sufficient to distinguish these three categories of urban cats, whose characteristics are summarized in Table 1.

When deliberating on the appropriate management of cats in urban environments, it is important to realize that the set of ecological, social (in relation to other cats and to humans), and behavioral needs of each group varies to a degree that entitles us to speak of different cat or cat-human cultures. The phrase “cat culture” was first used by sociologists Janet and Steve Alger in describing the environment of a cat shelter where humans and cats interacted and caretakers took into consideration the different temperaments and habits of individual cats in assessing their needs [24,25,26]. The concept of the social life and inner cognitive-affective world of cats as a culture can be understood theoretically from a biosemiotic perspective, which emphasizes the ability of animals to actively interpret their surroundings (the concept of Umwelt) [21,27,28]. This concept makes apparent the increased role of the social sphere in contrast to purely genetic dispositions, fully corresponding to the significant ecological plasticity of this species. Paul Leyhausen observed cats in Paris gathering in a single location without displaying the usual territorial aggression and described the tradition as “social gathering” [9]. The generational continuity of behavioral characteristics is exemplified by the fact that during a sensitive period (3–8 weeks of age), kittens adopt from their mothers the manner of relating to other cats and to people, and these early experiences have long-lasting effects into adulthood [29,30,31]. Mistrust of humans is transmitted intergenerationally in feral communities, while among pets the need for physical and social contact with people is an important component of their welfare.

While in the case of feral cats, their dependence on humans is indirect, and in the environments of Western cities, they are merely tolerated, pets are connected with humans to an extent that they cannot be considered a separate population. This is most evident with neutered individuals, who in some regions of the West make up the majority of the cat population, but do not contribute genetically to its future composition (in Shirley, Southampton, the estimate in 1994 was 96.8% of adult males and 98.7% of adult females; among all owned cats in the USA, the estimate was 79.8% in 1994 and 80% in 2007) [20,32,33]. In the case of pedigree cats, the appearance and behavioral characteristics of a breed are objects of intensive artificial selection. Conversely, we can say that people adapt to the needs of “their” cats in that they do not stay away from their homes for long periods. We can thus speak of the specific humano–cat culture of pets, whose existence is also reflected in the fact that domestic cats have gradually spread to all continents [21,34].

On a theoretical level, there arises the possibility that the need to regulate cat populations would apply exclusively to *feral cats*, who, because of their limited access to human-mediated diets, pose a greater threat to populations of birds and small mammals. Attempts to extinguish feral cat populations, such as in Australia, are a consequence of such considerations [35]. Concern for the welfare of cats would then be concentrated only on pets. This is problematic on two levels. In practice, it cannot be declared that ferals and cats living in close contact with humans are clearly divisible by group. If we see a tabby cat walking on a street, it is not clear to which category it belongs. This raises a considerable dilemma, as, depending on the categorization, the same individual can be seen as an *object of regulation* (feral) or of *concern for its welfare* (pet). On another level lies a distinct ethical problem: is it possible to give different normative valuations of members of the same species?

The majority of studies a priori perceive the very existence of feral colonies as a problem that needs to be solved by human intervention [4]. In recent decades there is increasing concern for the conservation of species that fall prey to roaming cats [8,36]. In addition to predation, problematic factors often named include disease transmission (to pets, livestock, wildlife, humans), noise during mating, and the presence of excrement in public spaces [23,37]. From a different perspective, one can assume that urban residents are averse to feral kittens’ high susceptibility to disease and high mortality (87.5% [38]).

Some studies are emerging that demonstrate an important function for ferals in the genetic continuity of cat populations [20] or demonstrate their affiliative relationships with particular people [10,11]. Members of animal rights organizations also have a positive relationship with feral cats in that they perceive them as objects of care, which mainly concerns “rescuing” feral kittens. Some supporters of animal rights do not acknowledge the independent right of reproduction, which in domesticated animals is often seen as perpetuating their suffering [39]. This attitude (common in urban shelters) is applied in relation to feral colonies, whose numbers are reduced in the name of preventing unnecessary suffering, through trap–neuter–return (TNR) programs or adoption (which is also associated with neutering) [40]. Such an approach is usually perceived as more humane than a direct eradication of feral colonies (but see [41,42]).

Even if we encounter a concern for welfare in the management of feral cat colonies, it must be realized that this is rather about choosing between the preferred outcomes of different groups of urban residents than a direct consideration of the interests of feral cats themselves. In theory, management decisions motivated by an authentic interest in the welfare of feral cats should have two stages: (i) what is in the interest of feral cats as an independent population/culture; (ii) how can these interests be reconciled with the needs of different groups of urban residents (who may also be representing the interests of other animal species)? In practice, however, stage (i) is rarely taken into account. Feral cats are regarded through the prism of welfare as fully domesticated animals (for whom a maximal life expectancy, for example, is considered desirable), or it is automatically assumed that numerical regulation of colonies is necessary and desirable. The significant dispute between proponents of the TNR method and of direct euthanasia of feral cats takes place in stage (ii): the decision is between which set of institutional actions is less stressful for feral individuals, which, however, does not mean that a given type of management is applied primarily in the interest of their welfare.

In all cases of feral cat management (euthanasia, TNR, transfer to cat shelters), a consistent implementation would lead to the disappearance of the specific behavioral-social manifestations of urban populations, which significantly mirror the general ecological strategies of domestic cats as a semi-wild species. These can come into conflict with certain *anthropocentric leanings in the evaluation of cat welfare*. As an example, from the perspective of the natural reproductive dynamics of feral populations, kittens are not pampered playthings; rather, their large litters represent an expendable resource strategy [43]. 

Assessing the welfare of *pets* is a difficult task. Unlike with feral cats, we cannot speak of a distinct population, as the movement, behavior, and reproduction of pets differ according the type of cohabitation they have with their owners. An individual cat’s quality of life is linked with the tolerance by members of the household of its behaviors, which may be manifestations of its own well-being but can be viewed negatively by humans. Steps taken in the supposed interest of pets, then, are in reality always a compromise of the interests of the individual actors (this should also include the point of view of species preyed upon) [44,45]. The theoretical and ethical dimensions of the problem are also unclear. On the one hand is a *utilitarian perfectionist stance*, subordinating the satisfaction of instinctive desires to overall quality of life (i.e., long lifespan) [18]. Such a view typically leads to neutering and confining pets. On the other hand is the traditional *laissez-faire policy*, which prioritizes the fulfilment of psychobiological needs of animals (e.g., engaging in predatory behaviors) [46]. The latter approach can combine intense care for an animal’s health (e.g., regular visits to the veterinarian) with the possibility of free movement, even though this can potentially bring harm. As research in several European countries has shown, such an approach is taken by most pet-owning households [19,45]. 

In the matter of pet welfare, the issue of neutering is a chapter unto itself. In contemporary Western society, there is a broad consensus supporting the neutering of cats that are not kept for breeding purposes [47,48]. Taking into consideration that “complications may develop from anesthesia or surgical trauma,” the main arguments for neutering are: (i) the prevention of potentially unwanted kittens; (ii) a reduction in behavioral problems in relation to owners or other people (e.g., increased aggression) [46]. It is important to realize, however, that this argument is not valid from a laissez-faire perspective, as it does not allow for certain key psychobiological needs of the animal associated with mating and nurturing offspring [16]. *Routine neutering* is particularly problematic, as it has the potential to significantly affect the population dynamics of the entire species.

From a population genetics and ecological perspective, the implementation of routine neutering constitutes disruptive selection: from free-ranging cats, only individuals who have learned to completely avoid humans and perceive them basically as predators remain intact. If they are living in groups, these represent a type of wild population whose members are difficult to redomesticate (classified as pseudo-wild in [20]). At the opposite end of the spectrum, then, are more and more individuals whose movement beyond the grounds of their owners is very limited (perhaps in a protected enclosure in the garden), and, regarding their possibilities of reproduction and social contact with other cats, they fall under full control of an owner. In light of this trend, it is necessary to observe that the dynamics of the human–cat relationship are shifting, at least in Western urban areas. At the same time, it is reasonable to suppose that the kind of interaction between humans and free-roaming cats who decide themselves whether to spend the night near a human dwelling or under cover of darkness has remained prevalent throughout the history of human–cat cohabitation, which goes back to ancient Egypt [13].

## 3. Threefold Modeling of Urban Cat-Human Relations

In understanding the multifarious interactions between humans and cats in urban environments, it is important to keep in mind that there are differences in opinion, not only on interspecies contact, but also on how to view cats themselves. The peak of conflict here are the “cat wars,” in which one side maintains that pet cats should be kept indoors or have restricted outdoor access, while the other side is of the opinion that companion cats should be allowed to move about in public spaces [3]. It would be naïve to believe that this disagreement is based on objective research into the ecological and social role of cats; it arises, first of all, from a change in society, which then sees cats through different eyes. It is important here to observe the shift of assessment criteria from economics or social utility to the problem of welfare. In the Anthropocene, we are witness to a widespread conviction that, in the interest of welfare, active interventions into the ecological relations or even the physiology of a given species are necessary. To an increasing degree, such efforts are propelled by institutions driven by both general demand and scientific studies. 

To better understand the complex dynamics of these relationships, it will be helpful to distinguish which cat populations are concerned. For each, we will note the mutual influence of three layers: *zoosemiotic interactions*, *institutionalized actions*, and *cultural representations*. Zoosemiotic interactions include intra- and inter-species contacts of an individual nature—for example, greeting rituals between cats or vocal communication between a cat and its owner. From a biosemiotic perspective, the cat and the owner here are both active agents whose situational behavior is modulated by individual experience. In contrast to this are institutionalized actions performed through a mediator who does not have an established relationship with the particular animal and is acting in a professional capacity. Institutions, such as animal shelters or veterinary organizations, regulate the movement and reproduction of urban cats, generally in accordance with public opinion. Given that different residents can have diametrically opposed assessments of the same cat population, we need to include the fluctuation of cultural representations in the model [21]. 

*Confined pets* are the source of least social conflict, as their owners have full control over their movement and reproduction. These cats are in compliance with the idea of responsible private ownership and do not come into contact with other people, such as neighbors. Their owners may doubt whether keeping them exclusively indoors is good for their physical condition, but in the case of intact pedigree females, breeders prefer the assurance of maintaining a genetically pure line. With males, the risk of injury from fighting or from passing cars is considered too high. Regard for a cat’s welfare and a possible need to plan its reproduction thus leads to a complete restriction of free movement, which has the side effect of confined cats posing no threat to populations of wild animals, other pets, livestock or humans (e.g., through a transmission of parasitic diseases) [37]. Thus, confined cats do not bother bird watchers, nature conservationists, hunters, and other interest groups [19]. We might add that indoor cat owners are the ideal customers of a large-scale industry serving cat needs—everything from canned food, to litter boxes, to scratching posts. It is the invention of nutritionally balanced cat food and the commercial availability of other pet products since the 1950s that has made it possible to keep cats exclusively indoors [1] (Figure 1). 

At the other end of the spectrum, here are *feral colonies*. These cats have a good number of human opponents, due not necessarily to personal experience, but to negative cultural representations, which are further strengthened by some ecological, epidemiological, and veterinary studies. Feral cats are portrayed as effective killers, carriers of parasites and disease, and disturbers of the peace (e.g., loud mating noises). These colonies are the primary focus of institutions of public health and veterinary medicine, frequently in cooperation with animal shelters. There are, of course, countries where this picture is more complicated (in Europe, primarily the Mediterranean countries). Groups of cats living in historical city centers are seen as part of the *genius loci*, and individual residents are friendly toward them, valuing the positive aspects of their presence. Natoli points to the antidepressant effect of colonies on the individuals who feed them (“*gattare*” in Italian), the educational effect on those interested in animals, and lastly the aesthetic effect (cats as “living decoration”) [18]. Another positive aspect is predation of urban rodent populations. In Italy, concern for animal welfare extends to feral cats, thanks to law no. 281 (enacted 1991). The prevailing interpretation is that the population of colonies should be regulated, yet individual animals should not be subjected to unnecessary stress. TNR method is usually employed. In sensitive cases, veterinary authorities work with people who have regular contact with colonies to capture less timid individuals [47]. The question of whether sterilization is compatible with a concern for welfare will be discussed below, but here we can note that members of animal rights organizations, as well as opponents of feral cats, can agree on a policy of reducing the number of colonies, but not completely eradicating them (the practice of TNR, with aid from volunteers, is widespread also in Austria, France, Portugal, Spain [14], and in parts of the USA as well [49]). 

It must be taken into account that shelters and veterinary organizations do not merely carry out the will of urban residents, but actively shape the discussion on welfare and themselves serve as a sort of “executive body” for managing free-roaming cats. Here we should point out the hybrid operation of cat shelters: they are hostile to feral colonies because of concerns over public hygiene or for species preyed upon by cats, while they see the wandering individual cat as an object of care. This inconsistency is sometimes rooted in legislation differentiating the approach to feral and owned cats (e.g., in the northern territory in Australia) [35]. During research in Estonia, all cat shelter managers I spoke with were convinced that a cat should have an owner, whose responsibility it is to “supervise” its movement to a certain degree. This conviction is unambiguously expressed in the shelters’ practice of microchipping their animals, which in some cities (e.g., Tallinn) is a legal obligation [21]. 

Shelters have effective public relations and present themselves in the media almost exclusively as places where abandoned cats find a home. The downside of their function of providing (temporary) homes to large numbers of animals is that their inhabitants are exposed to communicable diseases, to which stressed animals are particularly susceptible [50]. Municipal shelters, even in good faith, often cannot provide animals the conditions they need, and diseased or behaviorally problematic individuals are frequently euthanized, if only because of limited space. In deliberating on an ethical way of dealing with feral populations, one must face the sad reality that the direct eradication of colonies can be a better solution than the incessant suffering of their individual members in shelters (e.g., if confined in small cages), since their chances of adoption are minimal. TNR would seem to be a compromise here, as it allows the returned cats to continue their lives of freedom, although it also raises concerns about welfare, especially regarding difficulties of social continuity given the changes in psychical, hormonal, and immune function that can result from neutering [40,41] (Figure 2).

At this point, it is useful to show how an understanding of the specific needs of particular cat populations can help shelters that wish to consider the welfare of their animals as individuals [24]. Firstly, it must be recognized that representatives of different cat cultures have different social needs when it comes to the other cats and the people at the shelter (cf. Table 1). For feral cats, it is most beneficial if they can move freely around the space, allowing them to maintain contact with other individuals according to their particular preferences. Shelter employees should give these cats their space and let them initiate brief contact with people (excepting, of course, those in need of medical treatment). On the other hand, cats who had an owner can be very stressed in the presence of other cats, and as a rule, need contact with staff and visitors. These individuals have a good chance at adoption, but in the meantime, it is important to ensure that their living quarters offer a hiding place and that they have an appropriate degree of seclusion. Medium-sized cages are a good option, so long as shelter employees provide isolated individuals regular physical social contact [51]. If a cat is taken off the street, the problem arises of identifying which category it belongs to; this, however, can be solved relatively easily by observing its reaction to other cats and to people [52,53]. Trained volunteers play a key role here, as they do also in resocializing cats who were previously owned by humans, but then spent a long time in the streets. 

Relations of city dwellers to *free-roaming pets* are highly variable. First, we should note that while the categories of feral and confined cats are conceptually and practicably distinguishable, free-roaming pets do not constitute a group with distinct margins. This group includes neutered and intact cats, cats with apparent owners or those who frequent multiple households, cats who keep to themselves or who go on occasional wanderings. Such individuals are often the source of conflict between neighbors, which reflects the divided approach to free-roaming pets among the general public and among scientists as well. Concern for the welfare of pets can result in keeping cats indoors or allowing them to spend time both indoors and outdoors, and the decision of whether to neuter a cat depends primarily on the owner’s tolerance of the accompanying olfactory and behavioral manifestations. Solicitude towards cat welfare, then, can be seen rather as additional justification of the individual ideas of owners, as the arguments can go either way. This is increasingly compounded by concern for the welfare of species preyed upon by cats (especially birds), for which cat owners in different countries take varying degrees of responsibility [45]. Western city dwellers are generally coming to see the norm of neutering free-roaming pets as the responsible choice for owners and caregivers—it is endorsed by both “intolerant neighbor” types and influential organizations that see it as part of a comprehensive animal care fulfilling the criteria of welfare. 

As previously mentioned, neutering of males and females is a widespread practice with tremendous support from shelters and veterinary clinics. A routine part of this campaign is the portrayal of owners who do not neuter their cats as irresponsible (with the exception of pedigree owners); there are large numbers of cats in shelters, and every newborn kitten unnecessarily becomes a potential ward of these facilities. With increased public awareness and increases in neutering, however, situations can arise (e.g., in Finland) where the population of free-roaming cats significantly decreases and demand for adoptions must be met by shelters from abroad [21]. Here, we can see that the utilitarian perfectionist stance applied to the life of a particular animal is in direct conflict with the needs of the population of which it is a member. Added to this is the reality that the safety associated with keeping cats indoors has its downsides as well (boredom, obesity, stress), which further calls into question the veterinary conception of welfare [45]. A representative study of pets raised in Denmark has shown that while cats who are not allowed outdoors exhibit an increased degree of behavioral problems, pedigree cats are burdened with a higher incidence of disease [48] (Figure 3).

The appropriateness of neutering cats is not a purely scientific question, but to a significant degree also a valuative and sociocultural question. Various conceptions of animal welfare come into collision here—leaving aside the problem of interaction with other species, a positive evaluation of a neutered cat’s welfare stands or falls on the assumption that a potentially longer life is worth more than the hormonal, behavioral, and social processes associated with reproduction. The question of human responsibility for the behavior of cats is complicated by the fact that pet cats have many characteristics of a domestic species (confined pets being fully domestic), whereas feral cats, from a behavioral and ecological perspective, are at most a semi-domesticated species [13]. Added to this are the varying ethical views on the responsibility of humans for the reproductive scenarios of urban populations: a utilitarian perfectionist stance tends to assign full responsibility, while the laissez-faire approach sees cats as independent and free actors. One wonders whether *neutering campaigns*, at root, is not the reaction of a society that wants to eradicate the wild side of cat life (associated with the cycle of reproduction—instinctual freedom—death) as a projection of its own negative image [21]. We should recall that a condemnatory view of the fertility of cats is a repeating motif in history and played a role, for example, in the witchcraft trials of Tudor England [54]. 

Based on the examples of threefold modeling given here, it is difficult to avoid the concern that the one-size-fits-all approach to cat welfare supported by contemporary institutions (animal shelters, veterinary clinics) diverges from the actual psychobiological needs of many members of the species *Felis silvestris catus*. We have seen that the group of free-roaming pets is being exposed to pressure from the changing attitudes and needs of city dwellers. Even greater pressure is exerted on the culture of feral cats, which, to a significant degree, is intentional, given that they, unlike free-roaming pets, are portrayed in a thoroughly negative light. At the theoretical level, we need to subject to criticism the welfare concept for a species that most experts do not classify either as purely domesticated or as wild [55]. The guiding principle here must be an approach specific to the population and the individual.

## 4. Conclusions

In this study, I have highlighted three main problems of the cohabitation of humans and cats in urban environments: (i) interventions in feral cat colonies; (ii) keeping pets exclusively indoors; (iii) applying across-the-board neutering of all groups of cats except pedigrees. It is characteristic of the Anthropocene era that each of these steps is taken with reference to the declared welfare of individuals, while the specifics of particular cat cultures or cat-human cultures are forgotten [21,26]. From a population genetics perspective, consistent application of all three steps brings about a disruptive selection [20]. The problematic nature of this selection is aptly reflected in the question: “Is a world of sterilized feral cats and fertile cats of valued breeds desirable?” [23]. Natoli answers probably not, but adds that the successful sterilization of an entire urban feral cat population is unlikely anyway.

The number of free-roaming cats in urban environments is difficult to estimate. In the 1990s, the number of owned cats in the USA was estimated at 60 million and the number of feral and stray cats at 25–40 million [56]. In theory, data from municipal shelters could be a useful clue to population development, but these data can significantly vary among individual regions (on growth in Denmark between 2004 and 2017, see [57]; on decrease in ASPCA shelters in New York from 1934 to 1994, see [58]). In addition, a positive trend does not necessarily mean a growth in the population of free-roaming cats, especially in higher-income areas where residents “abandon” fewer cats on the street, but rather hand them over directly to the care of shelters [57]. Neutering campaigns are certainly rational in regions where the number of cats is generally considered too high—mainly in urban areas. From the veterinary view of cat welfare, neutering is positively evaluated based on the outcomes of a longer life span and a reduction in behavioral problems toward humans. 

We must realize, however, that neutering campaigns appear reasonable only because of their limited impact (they generally do not reach lower income households or rural communities) [19]. Otherwise, there would inevitably be an accelerated decline in the population of common short-haired pet cats. Owners who do not have their cats neutered are labeled as irresponsible by animal rights activists and by people who think there are too many cats in a given area (e.g., bird watchers, conservation advocates, or hunters) [59]. However, if neutering were to become an obligatory standard, this viewpoint would change: these same people would be providers of kittens, which (due to regulations) would become scarce commodities. From a global perspective, such a situation sounds like science fiction, but in some countries (England, Finland), it is becoming a local reality [20,21]. The laissez-faire approach to the question of pet reproduction is informed not only by a different assessment of welfare (e.g., the satisfaction of instinctive desires), but also by what type of cat is favored by the majority of people. In the end, it is the common cat owners who favor domestic short-haired cats that answer Natoli’s question in the negative. In the long term, it is preferable and in fact necessary that the respective proponents of the veterinary and laissez-faire views agree on a sensible approach to neutering cats, which in urban environments are neither a purely domestic species, nor a species independent of humans. 

The issue of keeping pets exclusively indoors is proving to be a case of value conflict. On the one hand is a *utilitarian perfectionist stance*, subordinating the satisfaction of instinctive desires to overall quality of life (i.e., long lifespan) [18]. On the other hand is a *laissez-faire approach*, which prioritizes the fulfilment of psychobiological needs of animals (e.g., engaging in predatory behaviors) [46]. While proponents of the first approach argue for keeping pet cats exclusively indoors, the other side points out that pets who are able to move about outdoors exhibit fewer behavioral problems. In this discussion, regard for the welfare of cats is confusingly combined with the issue of the welfare of species preyed upon by cats. This study argues that in the case of populations in European cities, and even in the USA to a significant degree, the negative effect of cat predation is probably overestimated. Through centuries of coexistence, the local fauna has had the opportunity to adapt, and it is likely that the abundance of resources associated with human presence has led to the increased density of both cat and bird populations in Western cities [6,7].

The issue of the cohabitation of humans and cats in urban areas is beset by conflicts between various interest groups and ideals of what the relationship between a cat and its owner should look like. Often forgotten is the diversity of the social life of cats and the values specific to each type of cat culture. We face the risk, then, that arguments on the welfare of cats are based on influential cultural representations rather than the interests of animals as members of groups distinguished as feral cats, free-roaming pets, and confined pets. Whether the wild side of cats is admired or perceived as a threat, people should accept it as a natural fixture of this semi-domesticated species. I am of the opinion that a greater awareness of the actual psychobiological needs of cats, including their reproduction, has the potential to clarify discussions on our mutual coexistence in urban spaces. Companion animal ecology should be developed, not only in close contact with veterinary and animal sciences, but should take into consideration the specific local situation, opinions, and habits of the individual participants in the debate [60]. Every city is a hybrid environment in which the mental worlds of humans (and their various interest groups) interact with those of cats and other domestic animals, as well as wild animals. The shape this coexistence takes is a matter of continual compromise, and the ideal of balance should be more important than the application of sweeping regulations tied to simplistic cultural representations that do not respect the complexity of ecological and social ties.

## Figures and Tables

**Figure 1 animals-11-00705-f001:**
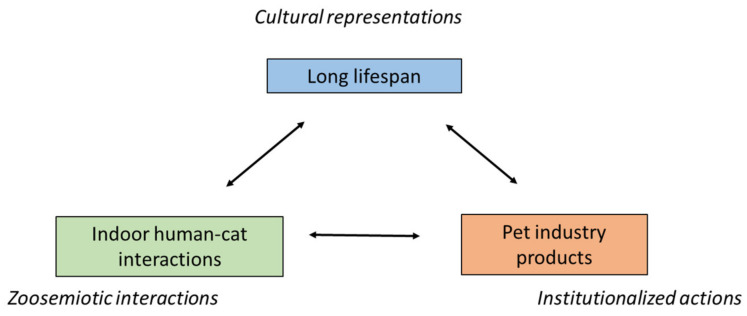
Threefold model of human–confined pet relations.

**Figure 2 animals-11-00705-f002:**
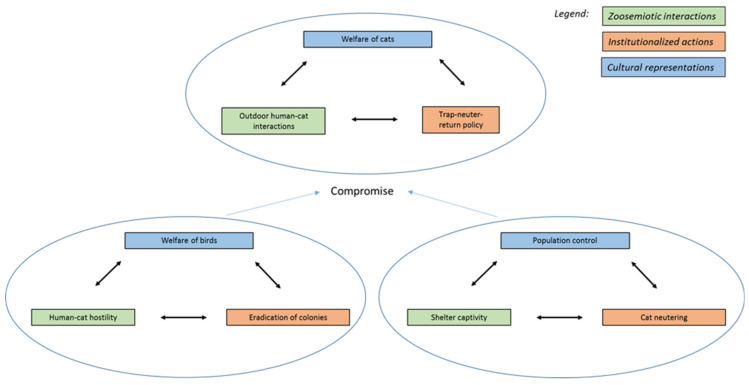
Threefold model of human–feral cat relations.

**Figure 3 animals-11-00705-f003:**
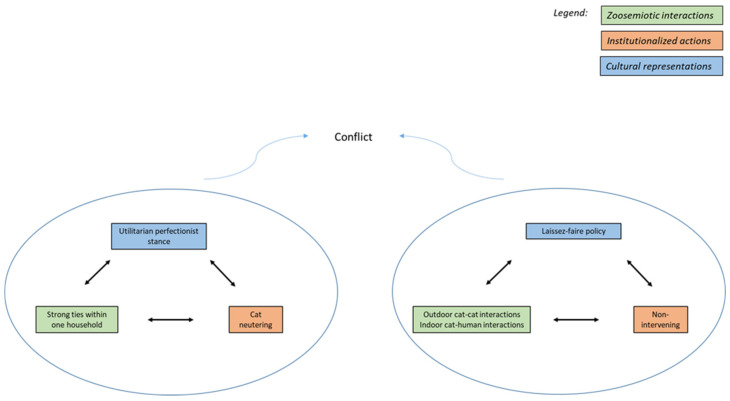
Threefold model of human–free-roaming pet relations.

**Table 1 animals-11-00705-t001:** Differentiation of three types of cat culture according to their relations to humans and other cats.

	Confined Pets	Free-Roaming Pets	Feral Cats
Reproduction	*Controlled*	*Regulated*	*Regulated or Uncontrolled*
Movement	*Controlled*	*Regulated*	*Uncontrolled*
Provisioning	*On a daily basis*	*Regular*	*Irregular*
Cat-cat socialization	*Limited*	*Territorially based*	*High*
Cat-human socialization	*High*	*Average*	*Low or None*

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
