# Peer review of "The Cohabitation of Humans and Urban Cats in the Anthropocene: The Clash of Welfare Concepts"

_animals, 2021, doi:10.3390/ani11030705_

Round 1

Reviewer 1 Report

Manuscript animals-1115841

Commentary

The Cohabitation of Humans and Urban Cats in the Anthropocene: The Clash of Welfare Concepts

By Filip Jaroš

The paper has been improved but I still have one comment: I disagree with author’s sentence: “In urban areas, Natoli’s categories b) and e) fall under same kind, hence my use of the term of feral cats is synonymous for 'unowned, free-roaming domestic cats'.” In fact, there is a misunderstanding. Categories b) and e) do not indicate cats of the same kind, since e) indicates cats that ‘after being domesticated, went back to a feral situation. They live on hunted prey; they are "independent from human beings" in regard to trophic resources’. Cats belonging to category b) ‘are completely dependent on food provided by human beings’. This is an important difference. Thus, if I were the author, I wouldn’t say that feral cats in the urban environment and in wilder environments like the mountains, where they live on hunted preys, belong to the same category. I know that also I utilised the term ‘feral’ to indicate unowned urban cats in the past, but now I would not, given that I realised that urban cats are different from the real feral cats.

DOMESTIC CATS (Felis silvestris catus)

Synanthropic cats

Domestic cats that live in the same environment as human beings. To this category belong:

a)    cats living exclusively at home

they are completely dependent on food provided by human beings and do not hunt

b)    unowned free-ranging urban cats

domestic cats that live freely in the urban environment. They are completely dependent on food provided either by human beings (cat-lovers) or found in the rubbish; they hunt very rarely

c)    category intermediate between (a) and (b)

domestic cats that have an owner but which live both inside and outside the home. Thus, they have contact with domestic cats belonging to category (b). They are completely dependent on food provided by humans (the owner, food provided by cat-lovers in the street and to remnants found in the rubbish)

d)    rural cats

domestic cats that live freely around a farm in the rural environment. They are "semi-dependent on human beings" (food provided by the owner and hunted prey)

e)    Feral cats

domestic cats that, after being domesticated, went back to a feral situation. They live in the same environment as wild cats. They live on hunted prey; thus, they can be considered "independent from human beings" in regard to trophic resources.

(Natoli, E.; Paviolo, M.; Piccoli, L. e Burla, P. 1996. Note sulla gestione delle popolazioni feline urbane. In: Linea Guida per l'Igiene Urbana Veterinaria, "Gestione delle popolazioni feline e canine in ambiente urbano", vol. II, ISS/WHO/FAO-CC/IZSTe/96.26(II). Teramo).

Line 198: “While the ecology and ethology of feral colonies remains an underresearched area…”. I do not know if the author is referring to the urban environment, but here there is a list of papers on the ecology and behaviour of urban cats, some on rural cats and some on real feral cats living on the sub-antarctic Kerguelen Islands:

  1. Natoli, E. 1985. Spacing pattern of a colony of urban stray cats (Felis catus) in the centre of Rome. APPLIED ANIMAL BEHAVIOUR SCIENCES 14: 289-304.
  2. Natoli, E. 1985. Behavioural responses of urban feral cats to different types of urine marks. BEHAVIOUR 94 (3/4): 234-243. doi: 10.1163/156853985X00208
  3. Natoli, E. 1990. Mating strategies in cats: a comparison of the role and importance of infanticide in domestic cats, Felis catus, and lions, Panthera leo L. ANIMAL BEHAVIOUR 40: 183-186.
  4. Natoli, E. & De Vito, E. 1991. Agonistic behaviour, dominance rank and copulatory success in a large multi-male feral cat, Felis catus, colony in central Rome. ANIMAL BEHAVIOUR 42: 227-241.
  5. Pontier, D. & Natoli, E. 1996. Male reproductive success in the domestic cat (Felis catus): a case history. BEHAVIORAL PROCESSES 37: 85-88.
  6. Natoli, E.; Ferrari, M.; Bolletti, E. & Pontier, D. 1999. Relationships between cat lovers and feral cats in Rome. ANTHROZOOS 12 (1): 16-23.
  7. Pontier, D. & Natoli, E. 1999. Infanticide in rural male cats (Felis catus) as a reproductive mating tactic? AGGRESSIVE BEHAVIOR 25: 445-449.
  8. Say L., Pontier D. & Natoli E. 1999. High variation in multiple paternity of domestic cats (Felis catus) in relation to environmental conditions. PROCEEDINGS OF THE ROYAL SOCIETY OF LONDON B 266: 2071-2074.
  9. Natoli, E.; De Vito, E. & Pontier, D. 2000. Mate choice in the domestic cat (Felis silvestris catus), AGGRESSIVE BEHAVIOR 26: 455-465.
  10. Natoli, E., Baggio, A. and Pontier, D. 2001. Male and female agonistic and affiliative relationships in a social group of farm cats (Felis catus). BEHAVIOURAL PROCESSES 53 (1-2): 137-143.
  11. Say L., Pontier D. & Natoli E. 2001. Influence of oestrus synchronization on male reproductive success in the domestic cat (Felis catus). PROCEEDINGS OF THE ROYAL SOCIETY OF LONDON B 268: 1049-1053.
  12. Pontier, D.; Say, L.; Debias, F.; Bried, J.; Thiolouse, J.; Micol, T. & Natoli, E. 2002. The diet of feral cats (Felis catus L.) at five sites on the Grande Terre, Kerguelen archipelago. POLAR BIOLOGY 25: 833-837.
  13. Say, L.; Devillard, S.; Natoli, E. & Pontier, D. 2002. The mating system of feral cats (Felis catus) in a sub-Antarctic environment. POLAR BIOLOGY 25: 838-842.
  14. Natoli, E.; Say, L.; Cafazzo, S.; Bonanni, R.; Schmid, M. & Pontier, D. 2005. Bold attitude makes male urban feral domestic cats more vulnerable to Feline Immunodeficiency Virus. NEUROSCIENCE & BIOBEHAVIORAL REVIEWS 29 (I): 151-157.
  15. Natoli E., Maragliano L., Cariola G., Faini A., Bonanni R., Cafazzo S. & Fantini C. 2006. Management of feral domestic cats in the urban environment of Rome (Italy). PREVENTIVE VETERINARY MEDECINE 77: 180-185.
  16. Natoli E., Schmid M., Say L. & Pontier D. 2007. Male reproductive success in a social group of urban feral cats (Felis catus). ETHOLOGY 113 (3): 283-289. doi: 10.1111/j.1439- 0310.2006.01320.x
  17. Bonanni R., Cafazzo S., Fantini C., Pontier D. & Natoli E. 2007. Feeding-order in an urban feral domestic cat colony: relationship to dominance rank, sex and age. ANIMAL BEHAVIOUR 74: 1369-1379.
  18. Cafazzo, S. & Natoli, E. 2009. The social function of tail up in the domestic cat (Felis silvestris catus). BEHAVIOURAL PROCESSES 80: 60-66.
  19. Pontier D., Fouchet D., Bahi-Jaber N., Poulet H., Guiserix M., Natoli E. & Sauvage F. When domestic cat (Felis silvestris catus) population structures interact with their viruses. Comptes Rendus, Biologies 332: 321–328.
  20. Martin J., Rey B., Pons J.-B., Natoli, E. and Pontier, D. 2013. Movements and space use of feral cats in Kerguelen archipelago: a pilot study with GPS data. POLAR BIOLOGY 36: 1531-1536. DOI: 1007/s.00300-013-1365-x
  21. Cafazzo S., Bonanni R. and Natoli E. 2019. Neutering effects on social behaviour of urban unowned free-roaming domestic cats. ANIMALS 9, 1105. doi:10.3390/ani9121105.

Line 378 and following: I am sorry, I suggested the wrong paper when I suggested to read the paper

Natoli E., Cariola G., Dall’Oglio G. and Valsecchi P. 2019. Considerations of ethical aspects of control strategies of unowned free-roaming dog populations and of the no-kill policy in Italy. JOURNAL OF APPLIED ANIMAL ETHICS RESEARCH 1: 216–229. https://doi.org/10.1163/25889567-12340014

I actually meant:

Natoli E., Ziegler N., Dufau A. and Pinto Teixeira M. 2019. Unowned free-roaming domestic cats: reflection of animal welfare and ethical aspects in animal laws in six European Countries. JOURNAL OF APPLIED ANIMAL ETHICS RESEARCH: 1-19. https://doi.org/10.1163/25889567-12340017

Reviewer 2 Report

This paper has been improved by better defining contained vs free ranging cats and by slightly expanding the list of impacts of cats to wildlife - although the omission to reference the increasing appreciation of the human and livestock health issues associated with free-ranging cats should be rectified.

There are a number of psychological/social terms used here that are not defined, nor are the costs and benefits of these terms defined. Eg an assumption is made that 'non-interventionist' approaches are ethically preferable , but surely this can not be justified for humans (medical treatment), or livestock, pets or wildlife that benefit from vet treatment. If the authors feel that intervening in an animal's life/welfare is necessarily a negative this should be stated up front and this will anchor the other arguments presented.

The authors have 'cherry-picked' scarce references suggesting that the health of contained cats is worse than free-ranging cats (when the reverse has been demonstrated and reported many many times) and also make the false statement that without intact free-ranging or feral cats there will not be a 'supply' of pet cats. This erroneous sentence likely resulted partly from a rewording of 'garden or stray' cats to 'pet' cats but the fact that euthansia and/or overcrowded cat shelters occur in most cities in most countries demonstrates that kitten production exceeds pet demand - and that habituated pet cats are best produced by breeders for the pet market. 

The premise of this paper seems to be that free-ranging cats are 'natural or naturalised' and desirable (for the cats and sometimes people) in some cities and that these populations are threatened by neutering. However the body of scientific evidence, repeated in this paper, is that neutering will rarely if ever eliminate stray and feral cat populations - which contradicts the premise.

This paper  is essentially an ideology and opinion piece rather than a scientific manuscript. Opinion pieces are  sometimes valuable and this Thesis could have value and may provoke some discussion but to do so it would be considerably enhanced by clearly stating a list of assumptions/ideologies, which are then explained or defended 

Ive annotated the manuscript with questions that I believe need to be addressed 

Reviewer 3 Report

I have three minor comments left:

1) You refer to a study of Peter Sandøe, 55, where we show a major increase in the number of cats coming into shelters.  This is in the context of discussing a potential increase in the number of stray cats. However, you fail to mention that we in the paper argue very strongly that the increase does NOT reflect an increase in the population of stray cats.

2) You several places argue that with current trends in cat management there is a real risk that there will be a shortage of non-pedigree cats. I think this claim is out of touch with reality. There is and will continue to be a large population of so-called "irresponsible" cat owners who have intact cats. They live in poor urban areas or in the country-side close to cities. There is no danger that this supply will dry out. And your claim that there is weakens your otherwise rather sane presentation.

3) Several places you indicate that feral cats enter shelters. They may in some countries, but I think it is rare. And as regards Denmark I can say with certainty that these cats when trapped are either (in most cases) euthanised or are put into TNR programs. I think a consequence of you view (which I sympathise with) is to accept a certain level of euthanasia of feral cats which for one reason or another become a nuisance. Saying that loud and clear will strengthen your argument.

Otherwise a fine piece of work that hopefully will generate a good discussion.

Author Response

This manuscript is a resubmission of an earlier submission. The following is a list of the peer review reports and author responses from that submission.

Round 1

Reviewer 1 Report

This is carefully researched, well argued may be the best I've had the opportunity to read in my 5+ years reviewing for Animals. It addresses a highly relevant issue, with implications for cats, people, and other species. It is grounded in cutting-edge literature. The modeling is clear and persuasive. The writing is excellent. I have no additional changes to request. 

Author Response

Thanks for a positive review!

Reviewer 2 Report

General comments

This is an interesting and very useful paper that highlights how the management of domestic cats in an urban environment poses problems of respect for the well-being of the animals themselves, which differ according to the cat categories; the latter are assigned to the categories according to their dependence on human trophic resources and their limitations posed by human beings. I think that it is worthwhile publishing this paper, but it must be powered increasing the clarity of some paragraphs, obscure at the moment. I will give some examples.

Line 78: be careful, you utilise here the words “pseudo-wild” for the first time. Although the meaning is clear, and you will give a definition later, maybe you should build a table to refer to, where there are all definitions. For example, here there are a table of categories of cats based on dependence/independence from food provided by human beings. You have not to utilise it necessarily, it is just an example.

DOMESTIC CATS (Felis silvestris catus)

Synanthropic cats

Domestic cats that live in the same environment as human beings. To this category belong:

a)    cats living exclusively at home

they are completely dependent on food provided by human beings and do not hunt

b)    unowned free-ranging urban cats

domestic cats that live freely in the urban environment. They are completely dependent on food provided either by human beings (cat-lovers) or found in the rubbish; they hunt very rarely

c)    category intermediate between (a) and (b)

domestic cats that have an owner but which live both inside and outside the home. Thus, they have contact with domestic cats belonging to category (b). They are completely dependent on food provided by humans (the owner, food provided by cat-lovers in the street and to remnants found in the rubbish)

d)    rural cats

domestic cats that live freely around a farm in the rural environment. They are "semi-dependent on human beings" (food provided by the owner and hunted prey)

e)    Feral cats

domestic cats that, after being domesticated, went back to a feral situation. They live in the same environment as wild cats. They live on hunted prey; thus, they can be considered "independent from human beings" in regard to trophic resources.

(Natoli, E.; Paviolo, M.; Piccoli, L. e Burla, P. 1996. Note sulla gestione delle popolazioni feline urbane. In: Linea Guida per l'Igiene Urbana Veterinaria, "Gestione delle popolazioni feline e canine in ambiente urbano", vol. II, ISS/WHO/FAO-CC/IZSTe/96.26(II). Teramo).

However, (this is just my personal opinion), I recently pondered that, since there are countless definitions proposed to define the unowned, free-roaming domestic cats (sometimes based on the environment where they live, or on their dependence/independence on human food resources and/or on their socialization with them), in my papers I decided to utilise the definition of 'unowned, free-roaming domestic cats', because it is the only one that all domestic cats that are not pets share, wherever they live, whatever they eat and in whatever way they relate to each other.

I don’t like Bradshaw categories (pseudo-wild, feral, semi-feral, pet, and pedigree) because I think that they add confusion to the picture, but I understand that you have to establish a point from which to define the categories utilised by you in your paper. And this is not very clear. I suggest to add a sentence after line 100 and before <<When deliberating on the appropriate management …>> like: <<In this paper, I consider the following cat categories: feral cats, garden cats and pedigree cats (Table …)>>.

Lines 130-135: <<This raises a considerable dilemma, as, depending on the categorization, the same individual can be seen … >> Well said!

Lines 175-176: <<…and relieve concern over the welfare of cats bred>>. What does the author mean here with “cats bred”? And why the responsible ownership would relieve concern over the welfare of cats bred? Please, explain better.

Lines 178-179: <<…subordinating the satisfaction of instinctive desires (of whom? Of cats?) to overall quality of life (of whom?)>>

Lines 181-190: I am quite confused. This is probably due to the differences in cultural background among different countries. Definitely, in Italy classical utilitarianism is not widespread among ordinary breeders but rather among ordinary people. In any case, the whole paragraph is not clear to me. Please, explain better.

Line 254 and following: to enrich the picture of cat colonies, you should read

Natoli E., Malandrucco L., Minati L., Verzichi S., Perino R., Longo L., Pontecorvo F. e Faini A. 2019. Evaluation of unowned domestic cat management in the urban environment of Rome after thirty years of implementation of the no-kill policy (National and Regional laws). FRONTIERS IN VETERINARY SCIENCE, section Veterinary Humanities and Social Sciences, 19 February 2019.

https://doi.org/10.3389/fvets.2019.00031

Natoli E., Cariola G., Dall’Oglio G. and Valsecchi P. 2019. Considerations of ethical aspects of control strategies of unowned free-roaming dog populations and of the no-kill policy in Italy JOURNAL OF APPLIED ANIMAL ETHICS RESEARCH 1: 216–229.

https://doi:10.1163/25889567-12340014

Minor points

I am not a native English speaker, but I think that some sentences in the paper are obscure. I give some examples.

Lines 59-62: I am not sure that I understood well the meaning of this sentence: <<Feral individuals correspond to traditional characteristics of domestic cats as semi-wild commensals whose important ecological and ethological functions, unlike fully domesticated animals, are not under human control>>. Do you recognize the same meaning in this reformulated sentence? <<Feral individuals can be described as cats with the traditional characteristics of domestic semi-wild commensal cats whose important ecological and ethological functions, unlike fully domesticated animals, are not under human control>>. If yes, I understood. If not, maybe the sentence has to be re-written because it is obscure in the present form.

Lines 63-64: <<… high the degree of entanglement between the two species may be, …>>. Which species?

In any case, again I am not sure that I understood well the meaning of the sentence: << However high the degree of entanglement between the two species may be, in the spirit of non-interventionist ethics, a human should not feel responsible for a single cat whose reproduction, sustenance and movement are to a considerable extent independent of human intentional acts>>. To me, it is obscure.

Lines 164-167: <<It can reasonably be said that these cats perceive members of the households they are primarily associated with as their reference group, analogous to the position of conspecifics in the social life of feral cats>>. In my opinion, this sentence is naïve. Tabor wrote a very nice booklet but many many years ago. Today, there are many scientific papers, on one side on cat social organization and, on the other side, on human-cat relationships, that do not support this view.

Lines 227-228: <<… three layers: cultural representation, institutionalized action, and zoosemiotic interaction>>. The following explanation should follow the same order. For the reader, it is easier to follow.

Reviewer 3 Report

see appended file

Reviewer 4 Report

This is an atypical piece for Animals. Not an empirical study, and not a review. More in the nature of a philosophical essay.

However, I like it and think that it makes a refreshing contribution to the ongoing discussion about cat management and cat welfare.

I have two main empirical concerns:

One is relating to the notion of a feral cat: The paper explicitly restricts itself to talk about cats in urban environments in Europe and North America. And I wonder if it is true that there are urban populations of cats living independently of humans? In in my group (situated in Denmark) we do research on unowned cats - not yet published. Our experience is that there are no cats living independently of humans. Even unsocialized cats are typically fed by humans (cat ladies and the like). The paper makes some references to feral cats in the Northern Territories in Australia. But these cats are not urban. So I guess that a bit more research on this should be done and that the result will be that there are no feral cats, but unowned, unsocialized cats which still are very dependent on humans. 

This conclusion, if you buy it, means that some of the ethical discussion should be re-written. We do not in urban settings have pure cat cultures but different cat-human cultures. And the discussion will be about tolerance towards these different cat-human cultures. This will also give rise to some interesting human-human ethical discussions since some of those humans looking after cat colonies are themselves marginalized.

The other concern is about the notion of pedigree cats. Together with colleagues I have done the only representative study of cat ownership: Sandøe P, Nørspang AP, Kondrup SV, Bjørnvad CR, Forkman B & Lund TB (2018). Roaming companion cats as potential causes of conflict and controversy – a representative questionnaire study of the Danish public. Anthrozoös 31(4): 459-473. As you will see here most owned cats are not pedigrees (purebred) and most are allowed outdoor access. So you should cite this and other evidence about how cats are kept and use it to modify your categories of cats.

When it comes to welfare problems of indoor cats you may also consider citing another (sorry!) of my papers which documents welfare problems for indoor cats: Sandøe P, Nørspang AP, Forkman B, Bjørnvad CR, Kondrup SV & Lund TB (2017): The burden of domestication: a representative study of welfare in privately owned cats in Denmark. Animal Welfare 26(1): 1-10. Here you will also find other references to support the claims you are making.

Also I think it would be a good idea for you to take a look into some to the great work done by Dennis Turner and his group, which serves to document that even undsocialized cats can bond to (individual) humans, while still being shy to humans in general. You may start here: Dennis Turner, A review of over three decades of research on cat-human andhuman-cat interactions and relationships, Behavioural Processes 141 (2017) 297–304.

So the main upshot of my comments is that you should modify your otherwise interesting analysis in light of current knowledge about different groups of urban cats and their behaviour.
